# Early Immune Response in Foreign Body Reaction Is Implant/Material Specific

**DOI:** 10.3390/ma15062195

**Published:** 2022-03-16

**Authors:** Nicolas Söhling, Muriel Ondreka, Kerstin Kontradowitz, Tobias Reichel, Ingo Marzi, Dirk Henrich

**Affiliations:** 1Department of Trauma, Hand and Reconstructive Surgery, Goethe University Frankfurt, 60590 Frankfurt am Main, Germany; fabiola97.mo@gmail.com (M.O.); kerstin.kontradowitz@em.uni-frankfurt.de (K.K.); marzi@trauma.uni-frankfurt.de (I.M.); d.henrich@trauma.uni-frankfurt.de (D.H.); 2Heraeus Medical GmbH, 61273 Wehrheim, Germany; tobias.reichel@heraeus.com

**Keywords:** PLGA, foreign body reaction, scaffold, bone tissue engineering

## Abstract

The design of novel biomaterials should directly influence the host-immune system and steer it towards high biocompatibility. To date, new implants/materials have been tested for biocompatibility in vitro in cell cultures and in vivo in animal models. The current methods do not reflect reality (cell cultures) or are very time-consuming and deliver results only after weeks (animal model). In this proof-of-concept study, the suitability of a Whole Blood Stimulation Assay (WBSA) in combination with a Protein Profiler Array (PPA), as a readily available and cost-effective screening tool, was investigated. Three different biomaterials based on poly(lactic-co-glycolic acid (PLGA), calcium sulphate/-carbonate (CS) and poly(methyl methacrylate) (PMMA) were exposed to native whole blood from three volunteers and subsequently screened with a PPA. Individual reproducible protein profiles could be detected for all three materials after 24 h of incubation. The most intense reaction resulted from the use of PLGA, followed by CS. If even marginal differences in implants can be reflected in protein profiles, the combination of WBSA and PPA could serve as an early biocompatibility screening tool in the development of novel biomaterials. This may also lead to a reduction in costs and the amount of animal testing required.

## 1. Introduction

Implants made of biomaterials have become an integral part of modern medicine. By definition, the implant directs the host reactions to the therapeutic target [1]. The ideal implant is therefore reactive but does not provoke rejection in terms of the foreign body reaction.

The task of developing these materials is complex because an organism always reacts with a response in the first instance. However, the intensity of the reaction determines whether a material is classified as biocompatible or incompatible.

Immediately after implantation, a differentiated host reaction to the foreign material begins. The host reaction essentially corresponds to the response to isolated tissue damage [2] but differs in the final phase. 

The initial implantation trauma is followed in the so-called adsorption phase by activation of the coagulation cascade with resulting hematoma formation. A complex, temporary protein and cell matrix is initially formed on the implant surface. Erythrocytes, fibrinogen and thrombocytes, which are representatives of the complement system and non-specific antibodies are the leading components [3]. As a result, the hematoma matrix leads to an acute immune response of the innate immune system with activation of the humoral (complement system) and cellular defence (granulocytes and mast cells) [4]. This phase is called the inflammation phase. The first cellular representatives are neutrophil granulocytes (polymorphonuclear leukocytes) and mast cells. In addition to phagocytosis of cell debris and apoptotic cells, neutrophil granulocytes release proinflammatory cytokines such as IL-1β, tumour necrosis factor-α (TNF-α), and interferon-γ (IFN-γ) [5]. These signalling molecules in particular have a significant influence on the differentiation of immigrated monocytes into proinflammatory macrophages. They are characterised by the release of IL-12 and IL-23 as well as high levels of inducible nitric oxide synthase and the pro-inflammatory cytokines IL-1β, IL-6, TNF-α, PDGF, GM-CSF and GSF [6] and set the course for a strong inflammatory reaction at the implantation site. Furthermore, tissue mast cells degranulate and release IL-4 and IL-13 in addition to histamine and fibrinogen [2]. While histamine and fibrinogen have a modulating influence on phagocyte recruitment and their adhesion to the foreign body surface [7], immigrated monocytes or macrophages differentiate into repair macrophages at high IL-4 concentrations [8]. Thus, they promote the formation of components of the extracellular matrix and produce high concentrations of anti-inflammatory IL-10 [9]. Pro-inflammatory macrophages dominate the first 48 h in the inflammatory phase with high inflammatory cytokine concentrations [10]. As inflammation progresses, the polarisation to tissue repair macrophages changes under the influence of IL-4. Without interfering foreign bodies, classical wound healing continues in the remodelling phase [8]. Acute inflammation with activated macrophages and T-cells subsides. This is not the case with an inserted implant or foreign body where insertion is followed by a more or less pronounced pathological-chronic inflammatory reaction with the formation of granulation tissue and giant foreign body cells around the foreign body [11].

According to current knowledge, the adsorption and inflammation phases play a central role in directing these subsequent reactions. The protein burst of this initial phase paves the way for subsequent cell recruitment, cell activation and cell polarisation [12]. Thus, implant-dependent modifications of the early host response due to specific implantation trauma, chemical material composition, degradation of products, surface conditions, porosity, etc. may have a significant influence on the fate of implants [10,13]. However, despite great efforts, only a few generally valid statements on the influence of implant characteristics on the immune response are possible so far [14,15,16,17]. One major obstacle is that comparing material compatibility studies is complicated. Investigation conditions and time periods are very heterogeneous. Standardised test conditions do not exist to date. Studies on the compatibility of biomaterials are primarily carried out on living organisms in animal models. Usually, the animals live with the material implanted for a matter of weeks. Periods of only a few days or hours and thus the early phases of the host reaction can only be investigated with great difficulty. In addition, the effort is considerable. While studies on isolated cell populations ‘in vitro’ also allow the investigation of immediate reactions and kinetics, they reflect only one component of the complex network [16]. However, they can only provide generally valid results to a limited extent.

A whole blood stimulation can offer an approximation to the very early inflammation phase with its hematoma formation and first host response. A native blood sample delivers the main cellular and humoral players for the adhesion and initial inflammation phase. With a suitable test, the very early response to a foreign material is assessable [18,19].

This ‘proof of concept’ study focuses on the first 24 h of a WBSA. The immune response to three different biomaterials is investigated. In addition to the PMMA bone cement mixed with Gentamicin (PALACOS^®^ R+G, Heraeus Medical GmbH, Wehrheim, Germany), which is used in the bone reconstructive Masquelet Technique, a CS scaffold comprising Gentamicin as well (*Herafill^®^*, Heraeus Medical GmbH, Wehrheim, Germany), which is a bioresorbable bone void filler with bone regeneration potential, and a PLGA microparticle scaffold with Gentamicin are tested using a commercially available human PPA allowing the detection and relative quantification of 105 different proteins.

The main question is whether the different materials already generate an individual immune response in the early phase of the host reaction. If this is the case, material-specific differences could be reflected in the detected proteome profiles. If correlations between material-specific proteome profiles and literature-based biocompatibility emerge, an application as a screening tool for the development of new implant materials is conceivable. Direct material comparisons would be possible.

## 2. Materials and Methods

In the present study, the immunological signature of a PLGA scaffold was investigated in the WBSA according to the scheme in Figure 1. It was compared to the immunological signature of a CS scaffold and PMMA bone cement. The positive control was LPS-stimulated whole blood, and the negative control was native whole blood. Additionally, the scaffold surfaces were examined and characterised by scanning electron microscope images (SEM).

### 2.1. Specimen Preparation

The material for the PLGA scaffold was a two-component system based on Gentamicin-loaded PLGA microparticles and a liquid component that, upon mixing, was able to form a porous scaffold. After 10 min, round (∅5 mm, 18 ± 3 mg) test specimens were produced with a skin punch from the 2 mm-thick layer. The test specimens were stored in a sterile container until use. As CS scaffolds flat (2 mm thick), sterile granules with a medium particle size of 5 mm in diameter were selected from Herafill granules (3–5 mm in diameter) under sterile conditions and used without further processing. PMMA bone cement specimens were prepared under sterile conditions. The PMMA bone cement was prepared according to the manufacturer’s instructions and applied to a sterile Teflon block in a 2 mm-thick layer. After 10 min, round (∅5 mm, 37 ± 2 mg) test specimens were produced with a skin punch. The specimens were stored in a sterile container until use.

### 2.2. SEM

To analyse the surface characteristics of the specimens, an untreated specimen of each material was assessed using a scanning electron microscope. The samples were sputtered with gold (3 × 60 s, Agar Sputter Coater, Agar Scientific, Ltd., Stansted, UK) and analysed using a Hitachi FE-SEM S4500 (Hitachi, Dusseldorf, Germany) with a voltage of 5 kV. The images were digitally recorded using the Digital Image Processing System 2.6 (Point Electronic, Halle, Germany) [20].

### 2.3. Whole Blood Stimulation Assay (WBSA)

For the WBSA, a volume of 100 µL (mm^3^) of each of the test specimens (PLGA scaffold, CS scaffold, PMMA bone cement scaffold) in 1.5 mL DMEM (Dulbecco’s Modified Eagle Medium, Invitrogen, Bleiswijk, Netherlands) was presented and mixed with 500 µL whole blood from healthy donors (n = 3: N.S., M.O., D.H. 2 male, 1 female, mean age 30 years). Blood was collected using lithium-heparin coated monovettes (Sarstedt, Nümbrecht, Germany). The blood collection was covered by Ethics Vote (project no. 89/19) of the Department of Medicine of the Goethe University. All donors signed informed consent. The preparations were incubated for 24 h at 37 °C and then the supernatant centrifuged for 15 min at 1100× *g* and 4 °C. The supernatants were aliquoted and stored at −80 °C until analysis.

### 2.4. Determination of the Immunological Signature

The semi-quantitative detection of the proteins in the supernatant was performed using the *Proteome Profiler Human XL Cytokine Array Kit* (RnD-Systems, Minneapolis, MN, USA) according to the manufacturer’s instructions (see Table 1 for the exact cytokine position on the array card). Membranes were photographed immediately after the addition of the chemiluminescence substrate provided with a kit using a Fusion Fx7 gel scanner equipped with a high dynamic range digital camera (Vilber Lourmat, Eberhardzell, Germany). The exposure time was 5 min for each membrane. The images were stored as 16-bit non-compressed TIFF-files using the software Fusion (Vilber Lourmat). The results were evaluated in a two-step process using the image processing software *ImageJ* (http://imagej.nih.gov/ij/) (accessed on 31 December 2021). First, strong and medium strong spots were automatically identified and analysed by the software (determination of optical density: size in pixel × mean grey value). Then, the remaining weak spots were marked manually and calculated. The manual marking of the spots was checked by a second experimenter. The results were normalised to the mean value of the reference spots (6 per array).

### 2.5. Statistics

The results were presented as box plots of the median in diagrams, as median and interquartile ranges (median/25% quartile/75% quartile) in the description of the results.

Statistical evaluation was performed using the Kruskal–Wallis test with Bonferroni–Holm post-hoc analysis. *p* < 0.05 is significant. Due to the small group size (n = 3 blood donors) and the relatively large number of comparison groups (n = 5 (Control, LPS, PMMA, CS, PLGA), it was also legitimate to consider explorative statistical significances (*p**), which bear the risk of false positive significance. These values are characterised by *p* < 0.05 before alpha correction. After correction, the values are higher. In the assessment, the risk of false positive results must be considered due to the missing alpha correction.

## 3. Results

### 3.1. SEM-Characterisation of Scaffold Surface

SEM analysis revealed clear differences in the fine structure of the surface of the test materials. The PLGA sample have a smooth and dense surface. Pores are not visible. The rather smooth surface is characterised by relatively large, undulating structures with a mean size of 124 µm ± 32 µm (SD). The surface of the CS sample is highly characterised by small, irregularly shaped structures (mean size 3.3 µm ± 2.4 µm (SD)) superimposed on the relatively fissured basic structure. No porous structures are visible.

The PMMA sample shows consistently closely spaced smooth, hemispherical structures with an average diameter of 17 µm ± 2 µm (SD). These structures are often superimposed by hemispherical substructures in the size order of 5.3 µm ± 1.6 µm (SD). In addition, pore-like structures in the range of 8.3 µm ± 1.6 µm (SD) can be found (Figure 2).

### 3.2. WBSA and PPA

A WBSA was performed to investigate the immunological potential of the materials. The proteome of the aliquoted supernatants was qualitatively and semi-quantitatively investigated with a PPA Kit according to Figure 1. An overview of all the measured parameters with the mean values from all three blood samples is given in Figure 3. In the following, statistically significant and explorative significant parameters are considered in detail. Non-significantly increased parameters are then presented and characterised in tabular form.

#### 3.2.1. Induced Proteins with Significant Differences

For six proteins, a statistically significant higher release after incubation with PLGA was apparent, compared to all other comparison groups (Figure 4 and Figure 5). These factors are Gro-1α, IL-5, IL-8, Lipocalin, M-CSF and MPO. Functionally, these proteins can be assigned to chemotaxis-inflammation (GRO-1alpha, IL-8), inflammation (IL-5; MPO), differentiation (M-CSF) and transport (Lipocalin) (Figure 4 and Figure 5; Table 2).

#### 3.2.2. Induced Proteins with Explorative Statistically Significant Differences

For other parameters, so-called explorative statistical significance was detected (Figure 6 and Figure 7). This is indicated when, in a direct pair comparison the *p*-value is <0.05, but, after alpha correction, it is higher. Since the group size was only n = 3 and the number of comparison groups was relatively high with n = 5, it is acceptable to consider these explorative significances as well. However, due to the lack of alpha-correction, there is a risk of false positive significance. The parameters are: IFN-γ, IGFBP-2, IGFBP-3, IL-1α, IL-17A, IL-24, Kallikrein, MIP-3a, PECAM (CD31), Pentraxin-3, PF-4, SHBG, VCAM-1. Functionally, these proteins can be assigned to immune activation, inflammation (IFN-γ, IL-17A, IL-24, Kallikrein), regulation (IGFBP-2, IGFBP-3, SHBG), chemotaxis (MIP-3a, PF-4), wound healing (IL-24), adhesion (PECAM (CD31), VCAM-1) and complement activation (Pentraxin-3) (Figure 6) (Table 3).

#### 3.2.3. Non-Specifically Increased Parameters

Non-specifically increased parameters without significant group differences and with their most important areas of influence are shown in Table 4. Table 5 indicates measured parameters that led to no, or a very low, detectable signal.

## 4. Discussion

There is consensus that the first implant–host reaction already has a significant influence on the fate of an implant [21,22]. In this sense, the course leading to the integration or rejection of an implant is already set [23]. Therefore, design of new bioimplants/biomaterials focuses on how implant properties direct the host response in a desired way. Despite a significant amount of effort, a systematic approach to this topic is still lacking. A major hurdle is the detection of changes, when scaffolds/materials are varied [24]. When evaluating the influence of biomaterials on the host organism, many publications limit themselves for practical reasons to the measurement of a few indicator proteins [25,26]. Essentially, these are indicators of acute inflammation and macrophage polarisation [17,25].

A system with high reproducibility and results that are quickly available is necessary. Furthermore, such a system should give a more comprehensive picture of the implant–host contact, which would also ideally provide an outlook on the biocompatibility of the implant [24].

This ‘proof-of-concept’ study is a first approach to this topic. The results showed that a combination of a WBSA with a protein profiling assay is suitable for determining an implant-characteristic proteome profile for the host response to a specific material. Here the bulk of released, first-line proteins—like receptors, hormones, cell activators, alarmins, proteins of adhesion, complement response and many more—were taken into account [21,27]. A more refined view of the early response was possible. Foreign material contact affects multiple arms and levels of host response. Each implant directs the responses in a characteristic way with good interindividual reproducibility. It seems that material differences can be mapped as proteome differences.

In order to contrast these differences in particular, three materials were investigated—all strongly differing chemically and structurally from each other. Two of those are used daily in the treatment of bone defects.

The scanning electron microscopy images reveal different surface topographies for the individual materials. The spectrum ranges from a smooth, slightly structured surface (PLGA) to a rough and clefty surface in CS, and a spherically structured surface in PMMA (Figure 2A–C). The visible structures span a broad range from 3–124 µm. Pore-like structures with a diameter of a few micrometres are also visible in images from PMMA.

In multiple studies to date, it has been shown that the surface roughness, in addition to the chemical structure, has a significant influence on the bioactivity of materials or on the response of the host organism [23,28,29]. Influences of structures and pores in the micrometre range but also in the nanometre range have been shown. The surface wettability is strongly influenced, and thus also the protein and platelet adhesion [30]. Hydrophilic surfaces show lower protein absorption and platelet adhesion than hydrophobic surfaces [31]. In this regard, the contact angles of a material can provide an initial prediction of the materials character. They are a measure of the hydrophilicity of a material surface or of the material and can be varied according to the surface morphology [32]. Contact angles of more than 90° characterise hydrophobic materials. For the materials used here, angles ranging from 76–119° (PMMA 76°; CS (Herafill) 119°; PLGA 77°–100°) can be found in the literature [32,33,34,35]. Thus, the lowest hydrophilicity can be derived for Herafill, followed by PLGA and PMMA. This is contrary to expectations. A mineral material such as calcium sulphate/calcium carbonate with its ionic structure should exhibit high hydrophilicity. However, the chemical structure or composition of the materials must be taken into account when evaluating the contact angles. Herafill is not a pure mineral material. Rather, a conglomerate of calcium sulphate, calcium carbonate, gentamicin and tripalmitate is present [10]. The tripalmitate shields the ionic structure and reduces hydrophilicity. Thus, higher protein adherence and platelet aggregation would be expected. It also affects the direct release of calcium [32].

PMMA has the lowest contact angle of all three materials at 76° and is thus still considered hydrophilic. Hydrophilic properties are associated with good biocompatibility [14]. However, it has also been reported that the water film formed on the surface impedes protein adhesion/platelet aggregation. An effect on cell adherence is possible. For example, Dimers et al. demonstrated a direct effect of freely accessible surface topography on macrophage and monocyte adherence and orientation and expression of proteins like IL-1β, TNF-α and IL-6 [36].

The PLGA scaffolds consist of Gentamicin-loaded PLGA microparticles, which were fused together at near body temperature by means of a process described already elsewhere [37]. It hydrolytically degrades into its constituents (lactic acid, glycolic acid as degradation end products) in aqueous solution over several days to weeks. In vivo, the degradation reaction, which is enzyme mediated, also occurs and proceeds at an accelerated rate. SEM imaging reveals a smooth but textured surface without significant pores. The contact angles determined in the literature are between 77 and 100° and thus indicate a rather hydrophilic character. In the course of material degradation, this contact angle may change. Protein adherence and platelet aggregation between the other two materials would be expected.

In contact with fresh blood, the PLGA-based material induced a significant increase in multiple inflammatory and/or chemotactic mediators compared to PMMA bone cement and the CS scaffold. More specifically, the expression pattern for PLGA suggests that monocytes and granulocytes in particular are activated, both of which represent the so-called first line of defence against invading pathogens. The chemokines GRO-1α and IL-8 are secreted by monocytes and neutrophilic granulocytes and have a strong chemotactic and activating effect on neutrophilic granulocytes [38,39,40]. The cytokine M-CSF is also released by monocytes and affects macrophages and monocytes in various ways, including by stimulating increased phagocytotic and chemotactic activity [41]. An essential component of the antimicrobial arsenal of neutrophil granulocytes is myeloperoxidase (MPO). This enzyme catalyses the conversion of chloride ions to hypochlorite, which has a strong antibacterial effect [42,43]. The high release of this enzyme indicates an activation of neutrophil granulocytes in the PLGA group. A further indication of a comprehensive activation of the granulocytes is IL-5, which is produced by activated mast cells, a subpopulation of granulocytes, among others [44].

Based on the mediator profile, it can also be assumed that different lymphocyte populations are activated by PLGA. The aforementioned IL-5 can also be released by activated T-helper cells. The mediator IFN-γ is released by certain T cell populations as well as by natural killer cells (NK cells), which are classified as mononuclear cells like lymphocytes [45,46]. IFN-γ is an effective messenger, which leads to a broad activation of various immune functions such as the activation of monocytes and enhancement of antigen presentation [47]. IL-24 can be produced by myeloid cells (monocytes, granulocytes) as well as by lymphoid cells. In addition to the broad cell-activating and pro-inflammatory effect, wound healing should be promoted [48]. The mediator Kallikrein should also be mentioned in this context. Its vasodilating effect could improve the blood flow and thus the supply in the wound/graft area [49]. The sequence of mutual cellular activations can only be assumed on the basis of the results. Short-lived initial alarmins such as TNF-α and IL-1β, as primarily secreted by monocytes, were detectable only in trace amounts, while secondary pro-inflammatory and/or chemotactic mediators of both myeloid and lymphoid origin were detectable. PLGA as one of the most frequently investigated polymers in tissue engineering and drug delivery is described as immune-stimulating on the one hand but also as a non-activating material on the other. To classify these contradictory results, the morphology of the PLGA and its degradation properties depending on its copolymer ratio, molar mass, porosity, etc. must be considered [50,51]. PLGA materials with a highly porous surface and high degradation rate, for instance, may elicit an increased immune-stimulating reaction. A few PLGA particles released from the surface can already lead to an intense immune reaction described above. Some recent work has shown increased IL-1β production due to activation of the NALP3 inflammasome in myeloid cells by PLGA microparticles. Furthermore, these findings suggest that PLGA microparticles have intrinsic immunogenic properties [52,53,54]. Waeckerle-Men et al. and Lewis et al., however, in contrast, demonstrated that PLGA microparticles have no obvious immunostimulatory properties and may even be immunosuppressive [52,54,55,56]. Hence, there might be a correlation between PLGA particle size and the degree of stimulation. For example, it has been demonstrated that PLGA microparticles are not phagocytised by macrophages with the same avidity as has been demonstrated for nanoparticles, but that they attach to the cell membrane and a stronger inflammatory stimulus after uptake results [54].

The CS scaffold, also loaded with gentamicin, generates a characteristic proteome profile after 24 h, with a significantly higher protein level for IGFB-3. Explorative significance (indicated by a direct pair comparison with a *p*-value < 0.05, but which is higher after alpha correction) in terms of higher protein levels compared to that for PMMA bone cement could be detected for FGF-19, IL-17a and V-CAM-1. For DPPIV, GRO-1α, IFN-γ, IL-5, IL-8, IL-24, MCS-F, MPO and SHBG levels were significantly lower compared to PLGA. From this, an immunomodulatory influence of the material can be detected, but, in particular, it indicates that the inflammation is significantly lower than with PLGA. Both materials decompose, according to the solubility product, in aqueous solution. The main difference between the materials is the degradation products. In the case of the CS scaffold, for example, ions are formed (Ca^2+^, SO_4_^2−^, CO_3_^2−^) that also occur physiologically. In high concentrations, they have an immunomodulatory effect [22]. The pH does not change upon absorption compared to the acidic pH of degrading PLGA. This may be an explanation for the good biocompatibility of the calcium sulphate scaffold due to only low local inflammation.

The PMMA bone cement used is a gentamicin loaded poly(methyl methacrylate)-based polymer. The test specimens have a rough but closed surface, as revealed by SEM. Together with the low polarity of the surface, this makes it difficult for proteins to adhere to it and thus also for cells to adhere. Under physiological conditions, the material is almost inert, so that phagocytosis by macrophages is not possible practically speaking. The result is a pronounced foreign body reaction with the formation of a granulation membrane [57,58]. The WBSA shows a characteristic protein profile with significantly lower protein concentrations of ENA-78 (CXCL5), FGF-19, Gro-1α, IFN-γ, IL-5, IL-8, IL-17a, IL-24, Lipocalin, MCSF, MIP-3a, MPO, SHBG and VCAM-1. There is activation of all systems in the WBSA, but activation is much milder. In contrast to this, PMMA-microparticles demonstrated pro-inflammatory behaviour in vitro with isolated fibroblastic or monocytic cells. IL-1, IL-6, MIP-1, MCP-1 and TNF-α were released within the first 24 h [59,60]. In the course, this then turns into chronic inflammation due to a shift of macrophage polarisation towards the inflammatory M1 subtype.

Gentamicin was added to all three scaffold materials, but the release rate of gentamicin from the scaffolds is not known. Due to the structured surface of the PLGA scaffold and the CS scaffold, a higher release is expected. Thus, gentamicin could also contribute to the enhanced inflammatory response in both porous scaffolds. Indeed, Nau et al. found evidence for an enhanced immune response after implantation of bone cements containing gentamicin [58]. However, this cannot be determined conclusively and needs to be clarified in further experiments.

A major influence on the adherence of platelets and cells in the context of the innate immune response is fibrin adhesion. This is significantly influenced by the surface topography in the nanometre range. Highly structured surfaces tend to have a detrimental effect [61]. Here, the epitopes for cell adhesion, complement activation and coagulation cascade cannot unfold sufficiently. Smooth surfaces have been shown to be most effective for sufficient epitope presentation [30]. However, the impact on this study should be considered minor. The blood used here was fully heparinized. The coagulation cascade was thus switched off. The influence on platelets was thus also reduced [31].

However, a few major limitations are apparent. Native blood samples were obtained from three individuals to level out interindividual differences. Libers et al. reported a high interindividual variance in results for samples from different subjects [18]. However, valid results can be obtained by averaging the results of several interindividual samples. For intraindividual samples, Wouters et al. demonstrated only a slight variation [62]. For this reason, only one WBSA was performed per blood donation.

In principle, activation of the immune response and coagulation cascade naturally occurs through interaction with the blood collection system and the reaction container. However, this is true for all materials tested, resulting in a systematic error. It is advantageous, especially when using the WBSA, for the immune response to be viewed in isolation from implantation trauma. In this way, even subtle material differences could possibly be detected. Finally, it must be mentioned that the hematoma studied here then corresponds more to a tissue hematoma and not a fracture hematoma [63]. Conclusions must be drawn with caution.

The final influence of protein adherence and platelet aggregation on the foreign body response has not been clarified [5]. In principle, protein and platelet aggregation assays can, of course, provide information on this, but this is not the subject of this study. Instead, the aim here is to verify whether a WBSA combined with a proteome profiling array is suitable for the initial characterisation of biomaterials. In particular, the simplicity of the implementation is the main focus. If clear differences in the initial foreign body response emerge under these conditions, the concept can be used as a screening tool. The results presented here are encouraging.

Thus, finally, the WBSA offers an opportunity to simulate immunological responses on first implant-host contact in situ, as would be the case in a potential therapeutic use of the material [25,26,64]. The proteome profiling assay then allows the qualitative and semi-quantitative detection of a large proportion of the released proteins [16].

## 5. Conclusions and Outlook

Native whole blood stimulated with biomaterials allow a first host response to be measured with the releases of a material-dependent, highly specific proteome.

It now remains to clarify whether marginal implant differences are also reflected in a specific proteome profile and whether the proteome profiles allow a statement about material compatibility in vivo in the long term. If both requirements are met, the use of the WBSA in combination with a proteome profiling assay as a screening tool in biomaterial development is conceivable. Databases of protein profiles, similar to a fingerprint index, could be created. A mass of materials could then be pre-screened and characterised at an early stage using this rapid and cost-effective method [60]. Only a promising selection would then be tested in expensive animal experiments. This would allow costs and animal testing to be reduced.

## Figures and Tables

**Figure 1 materials-15-02195-f001:**
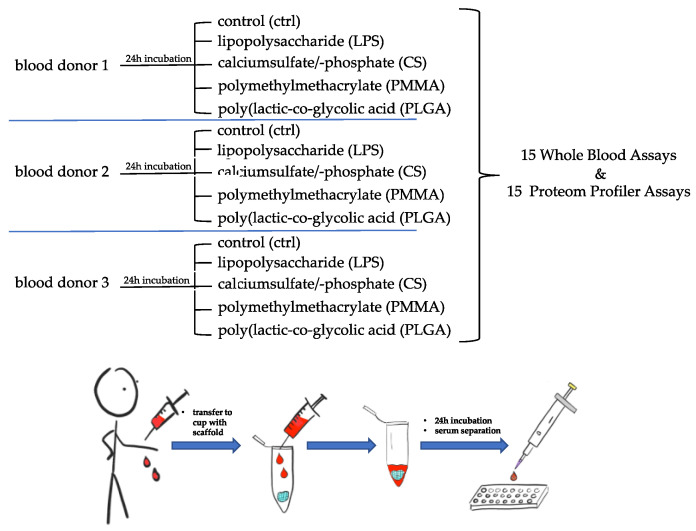
Schematic sequence of the experiments. Heparinized blood samples were not pooled. Scaffolds were presented in medium (DMEM) and blood was added in a 3:1 ratio (medium/blood).

**Figure 2 materials-15-02195-f002:**
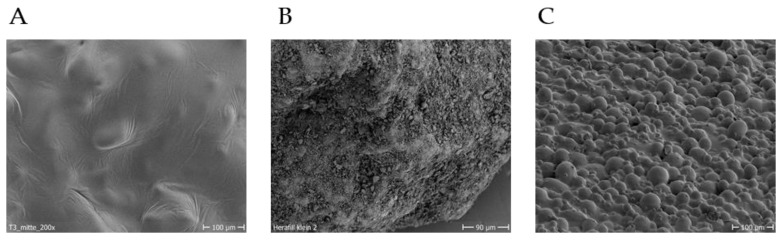
SEM images of the individual material surfaces. Different surface topographies of the individual materials are revealed. (**A**) poly(lactic-co-glycolic acid; (**B**) calcium sulphate/phosphate (Herafill); (**C**) polymethyl methacrylate.

**Figure 3 materials-15-02195-f003:**
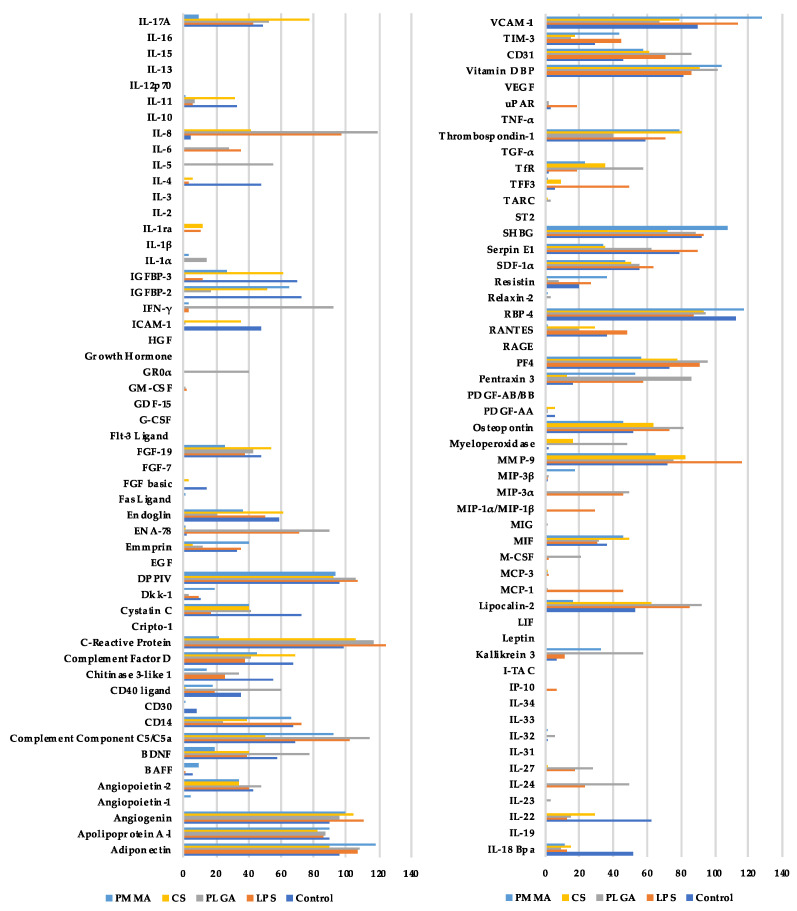
Overview of all the parameters measured. Median values of the three probands are presented. Ctrl (control), PLGA (poly(lactic-co-glycolic acid), CS (calcium sulphate scaffold), PMMA (polymethylmethacrylate).

**Figure 4 materials-15-02195-f004:**
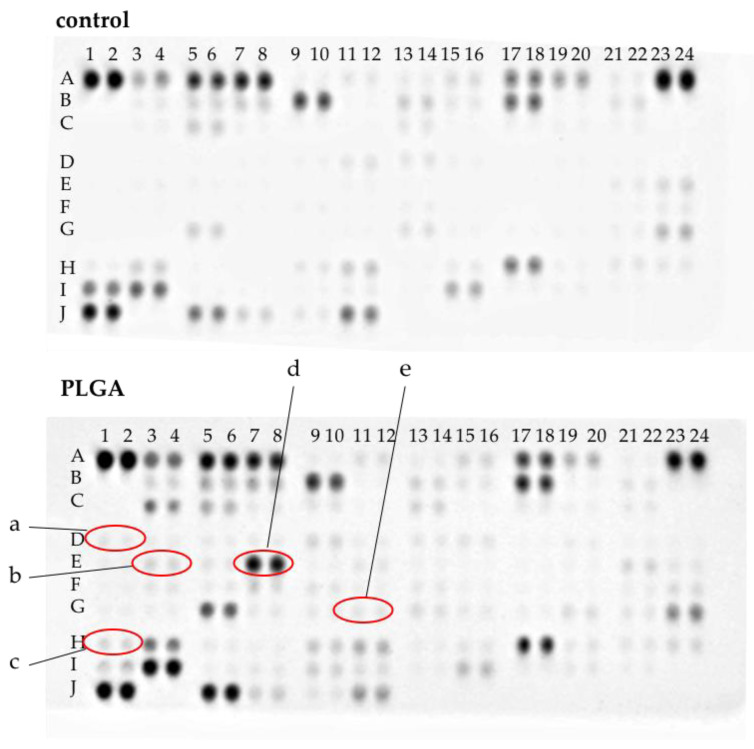
Representative protein profiler arrays for whole blood incubated with either medium (negative control) or PLGA. Proteins with a significantly increased concentration compared to control were marked. (a): GROα (D1–2), (b): IL-5 (E3–4), (c): IL-8 (E7–8), (d): M-CSF, (e): MPO (H1–2).

**Figure 5 materials-15-02195-f005:**
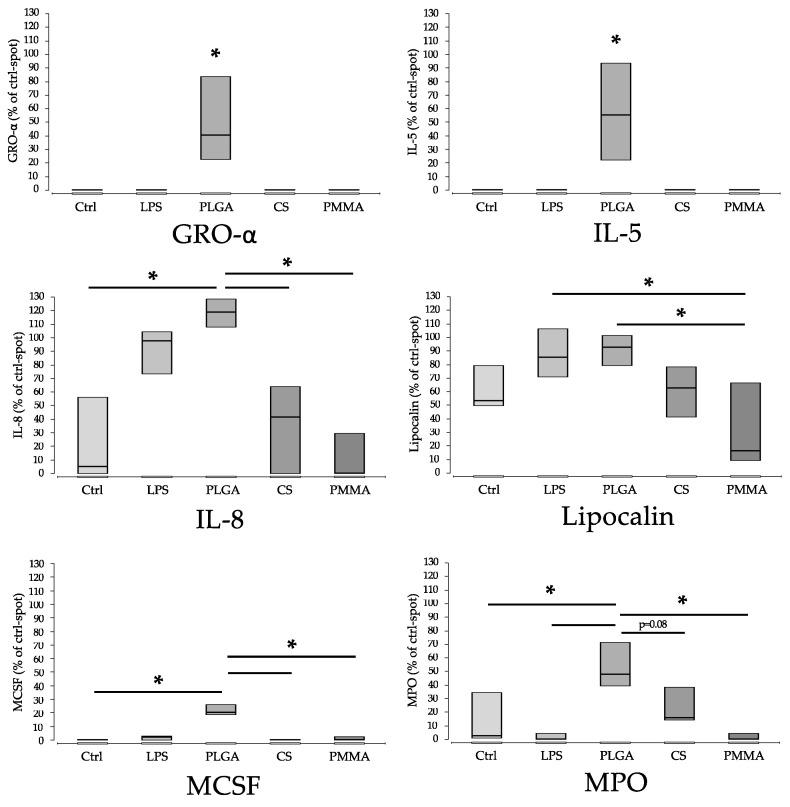
The sample materials induced significantly different proteins during WBSA. The values are normalised to the density of the ‘control’ spots (ctrl) (=100%). For PLGA significantly elevated protein concentrations were found for Gro-α, IL-5, IL-8, lipocalin, M-CSF and MPO. LPS alone shows comparable high values for IL-8 and lipocalin. ‘*’ marks significant differences (*p* < 0.05). CS, PMMA. (Ctrl: control; LPS: Lipopolysaccharide; PLGA: poly(lactic-co-glycolic acid; CS: calcium sulphate scaffold; PMMA: polymethylmethacrylate).

**Figure 6 materials-15-02195-f006:**
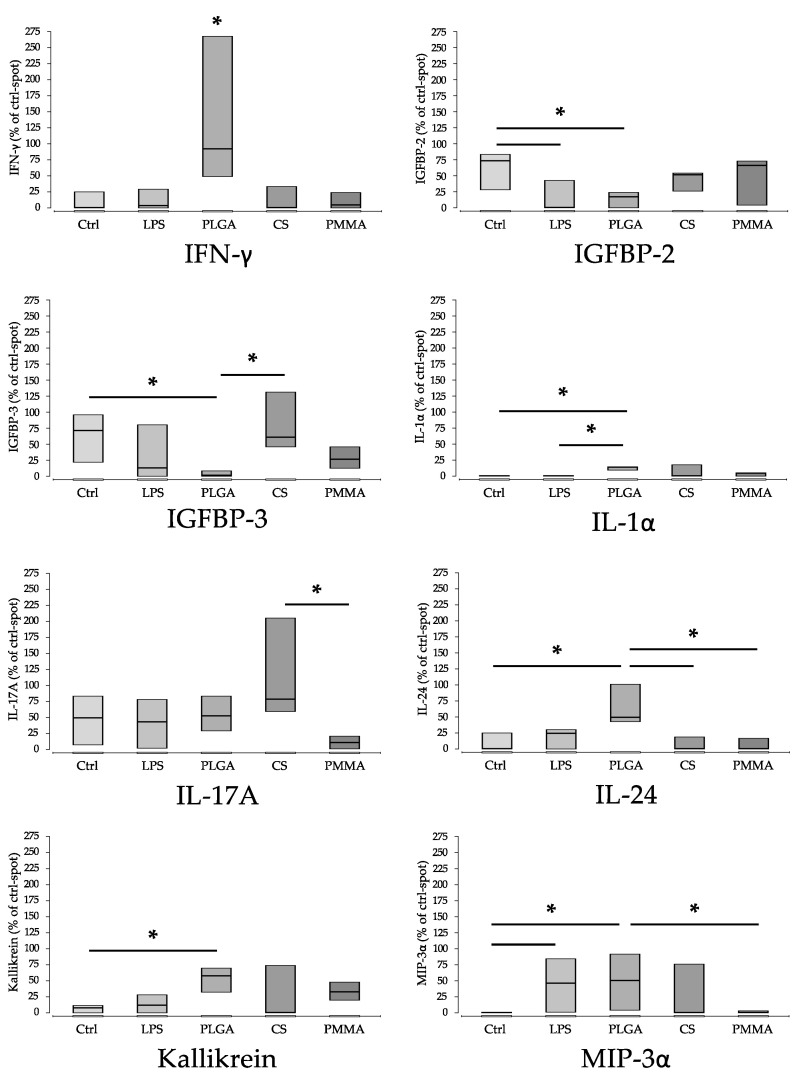
Parameters with explorative significant differences in non-parametric pair comparison without α correction (part 1). There is a risk of false positive significances. ‘*’ indicates explorative significance, *p* * < 0.05. Ctrl (control), PLGA (poly(lactic-co-glycolic acid), CS (calcium sulphate scaffold), PMMA (polymethylmethacrylate). (Ctrl: control; LPS: Lipopolysaccharide; PLGA: poly(lactic-co-glycolic acid; CS: calcium sulphate scaffold; PMMA: polymethylmethacrylate).

**Figure 7 materials-15-02195-f007:**
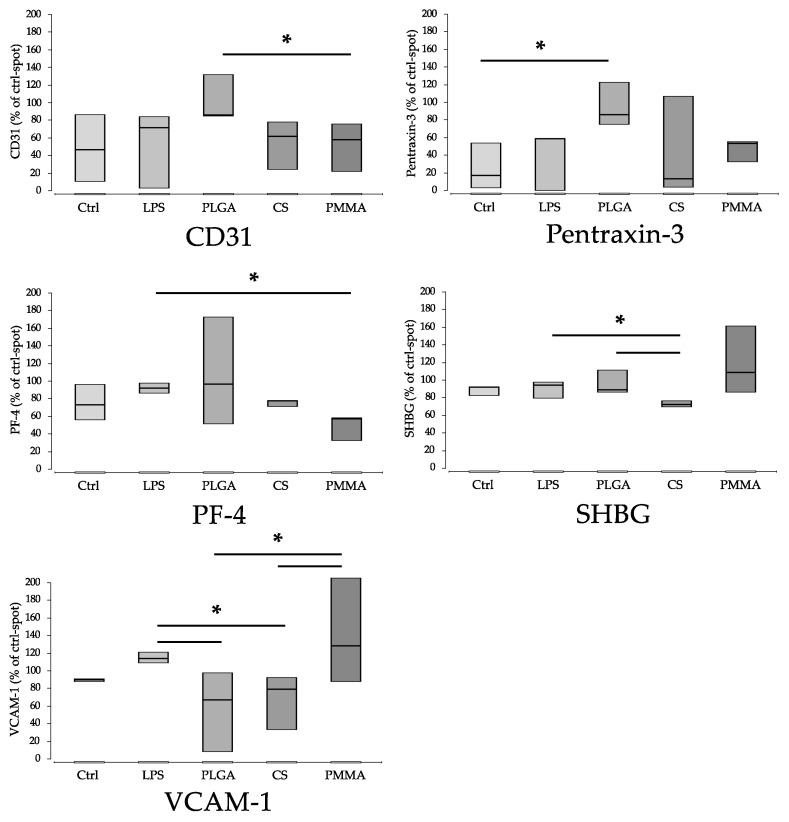
Parameters with explorative significant differences in non-parametric pair comparison without α correction (part 2). There is a risk of false positive significances. ‘*’ indicates *p* * < 0.05. Ctrl (control), PLGA (poly(lactic-co-glycolic acid), CS (calcium sulphate scaffold), PMMA (polymethylmethacrylate). (Ctrl: control; LPS: Lipopolysaccharide; PLGA: poly(lactic-co-glycolic acid; CS: calcium sulphate scaffold; PMMA: polymethylmethacrylate).

**Table 1 materials-15-02195-t001:** Illustration of each cytokine position on the Proteome Profiling Array sheet.

	1–2	3–4	5–6	7–8	9–10	11–12	13–14	15–16	17–18	19–20	21–22	23–24
A	-	Adipo-nectin	Apolipo-protein A-l	Angiogenin	Angio-poietin-1	Angio-poietin-2	BAFF	BDNF	Complement Comp C5/C5a	CD14	CD30	-
B	-	CD40 ligand	Chitinase 3-like 1	Complement Factor D	C-Reactive Protein	Cripto-1	Cystatin C	Dkk-1	DPPIV	EGF	Emm-prin	-
C	-	ENA-78	Endoglin	Fas Ligand	FGF basic	FGF-7	FGF-19	Flt-3 Ligand	G-CSF	GDF-15	GM-CSF	-
D	GR0α	Growth Hormone	HGF	ICAM-1	IFN-γ	IGFBP-2	IGFBP-3	IL-1α	IL-1β	IL-1ra	IL-2	IL-3
E	IL-4	IL-5	IL-6	IL-8	IL-10	IL-11	IL-12 p70	IL-13	IL-15	IL-16	IL-17A	IL-18 Bpa
F	IL-19	IL-22	IL-23	IL-24	IL-27	IL-31	IL-32	IL-33	IL-34	IP-10	I-TAC	Kalli-krein 3
G	Leptin	LIF	Lipocalin-2	MCP-1	MCP-3	M-CSF	MIF	MIG	MIP-1α /MIP-1β	MIP-3α	MIP-3β	MMP-9
H	Myelo-peroxidase	Osteo-pontin	PDGF-AA	PDGF-AB/BB	Pentraxin 3	PF4	RAGE	RANTES	RBP-4	Relaxin-2	Resistin	SDF-1α
I	Serpin E1	SHBG	ST2	TARC	TFF3	TfR	TGF-α	Thrombo-spondin-1	TNF-α	uPAR	VEGF	-
J	-	-	Vitamin D BP	CD31	TIM-3	VCAM-1	-	-	-	-	-	-

**Table 2 materials-15-02195-t002:** Expressed parameters with significant differences between groups and assignment to functional categories.

Parameter	Function
GRO-1 α, IL-8	chemotaxis-inflammation
IL-5; MPO	inflammation
M-CSF	differentiation
lipocalin	transport

**Table 3 materials-15-02195-t003:** Expressed parameters with explorative significant differences between groups and assignment to functional categories.

Parameter	Function
IFN-γ, IL-17A, IL-24, Kallikrein	immune activation, inflammation
IGFBP-2, IGFBP-3, SHBG	regulation
MIP-3a, PF-4	chemotaxis
IL-24	wound healing
PECAM (CD31), VCAM-1	adhesion
Pentraxin-3	complement activation

**Table 4 materials-15-02195-t004:** Expressed parameters without significant differences between groups and assignment to functional categories.

Parameter	Function
IL-1β, TNF-α, Chitinase 3-like 1, CRP, GDF-15, IL-2, IL-4, IL-6, IL-18, IL-23; IL-27, IL-32, MIF, MIG, Osteopontin	proinflammatory
IL-1RA, IL-10, ICAM-1, IL-27	anti-inflammatory
CXCL5(ENA78), RANTES (CCL5), SDF-1a (CXCL12), C5a, IP10, MCP-1, MCP-3, MIP-1a, MIP-3b, PDGF-AA/AB, TARC	chemotaxis
Fas-ligand (CD95L)	apoptosis
BAFF, DKK-1, Endoglin, GDF-15, GM-CSF, IL-11	differentiation
Angiogenin, Angiopoietin-1, Angiopoietin-2, Endoglin, IP10, PDGF-AA, Thrombospondin-1	angiogenesis
Osteopontin	osteogenesis
FGF basic, FGF-7, FGF-19, BDNF, Chitinase 3-like 1, EGF, GDF-15, GM-CSF, IL-4, IL-22,	Cell-growth and -proliferation, regeneration
C5a, Complement-Factor D, CRP	complement reaction
Serpin E1	coagulation
Emmprin, ICAM-1, Thrombospondin-1	adhesion
Adiponectin, Relaxin-2, Resistin	hormone
Complement-Factor D, Cystatin C, DPPIV, MMP-9	proteaseactivator/-inhibitor
Apolipoprotein A-1, CD14, Emmprin, RBP4, Vitamin D BP	molecule transport
CD14, CD30, IL-1RA, CD95L, CD40L, ICAM-1, TfR	soluble receptor
CD30, CD40L, BAFF, PDGF-AA/AB, ST2, TIM-3	cell activation
TFF3	still unknown

**Table 5 materials-15-02195-t005:** Measured parameters with marginal or undetectable expression and assignment to categories.

Parameter	Function
G-CSF, IL-13, IL-31, RAGE, TNF-α	pro inflammatory
IL-19, IL-33	anti-inflammatory
IL-16, I-TAC	chemotaxis
TNF-a	apoptosis
FLT3-Ligand, IL12p70, LIF, TGF-α	differentiation
VEGF	angiogenesis
FGF-7, FLT3-Ligand, G-CSF, HGF, IL-3, IL-15, TGF-α	Cell-growth and -proliferation, regeneration
Growth Hormone, Leptin	hormone
Cripto-1, RAGE	soluble receptor
HGF, IL12p70, IL-13, IL-16, IL-33, IL-34	cell activation

## Data Availability

For further data sets, please contact the corresponding author.

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
