# Peer review of "Early Immune Response in Foreign Body Reaction Is Implant/Material Specific"

_materials, 2022, doi:10.3390/ma15062195_

Round 1

Reviewer 1 Report

The authors examines the initial immune response on the materials, and in that respect, the paper has some value. However, on the other hand, it simply tests the immunoreactivity because there is no data of materials like morphology, contact angle, protein adsorption, etc.. In addition, the caption of Fig. 1 is not sufficient, and the numbers 1-24 and the test samples should be indicated. This paper is valuable as a report and as a test of the immune response of other implant materials, but the discussion and interpretation of the results are inadequate because of the lack of multifaceted data on the materials. The following corrections should be made to the acceptance.

1.P3 Materials and Method

WBSA: a volume of 100 micro-l of the respective test specimens 100 micro-g ?

2.Caption of Fig 1 should be redesigned for clarity. 3.

3.Other properties of the material (morphology, contact angle, protein adsorption data, etc.) should be added to the discussion.

Author Response

Reviewer 1

Response to reviewer:

We thank the reviewer for the intensive examination of our manuscript and the constructive criticism. Below we comment on the points of criticism. Since the reviewer requested a linguistic revision, the manuscript was revised by a professional Proof Reading Service (Proofers, GB) before resubmission.

1.P3 Materials and Method

WBSA: a volume of 100 micro-l of the respective test specimens 100 micro-g ?

Answer:          We changed this inaccuracy to “mm3/µL”.

2.Caption of Fig 1 should be redesigned for clarity. 3.

Answer:          We agree with the reviewers critism and added a table with cytokines position on array sheet for more clarity.

3.Other properties of the material (morphology, contact angle, protein adsorption data, etc.) should be added to the discussion.

Answer:           We thank the reviewers for the valuable comment. A corresponding section has been added to the discussion (p.12)

Reviewer 2 Report

This manuscript by Nicolas Söhling and colleagues reports two biocompatibility screening methods, Whole Blood Stimulation Assay (WBSA) and Protein Profiler Array (PPA), to evaluate the early immune response of implanted biomaterials. They investigated the protein profiles on three commonly used biomaterials PLGA, CS and PMMA by the two methods and indicated PLGA shows the most intense reaction. The authors’ idea is not very new but still informative, and their developed methods could be complemented with other biocompatibility evaluation tools for the development of new blood-contacting biomaterials. The manuscript may be eventually published after addressing the following concerns.

  1. The photograph and scanning electron microscope (SEM) images of the tested materials should be provided to show their morphologies, which might affect the protein adsorption on the materials’ surface.
  2. Platelet adhesion is considered as an early event when biomaterials come into contact with blood, and the adhered platelets could induce the following immune responses. I recommend the authors add the platelet adhesion assay on the tested materials.
  3. The references related to protein adsorption and platelet-surface interaction on biomaterials should be added and discussed (ACS Appl. Bio Mater. 2020, 3, 5574; Acta Biomaterialia, 2017, 64, 187; Biomaterials, 2010, 31, 1533)
  4. To my knowledge, pH 7.4 PBS or HBSS are usually used as buffer solutions in blood biocompatibility evaluation, but in the WBSA experiment the authors used DMEM solutions, a medium for cell culture. Please specify the reason for this.
  5. Figure 1, please add the description of letter ‘A-J’ and number ‘1-24’ in the figure caption.
  6. The information of Figure 4 and Figure 5 are not included in the Results and Discussion, please revise it.

Minor

  1. Please keep the range and scale of Y axis consistent in each figure.
  2. Line 337, revise ‘WBA’ to ‘WBSA’.

Author Response

Reviewer 2

Response to reviewer:

We thank the reviewer for the intensive examination of our manuscript and the constructive criticism. Below we comment on the points of criticism. Since one reviewer requested a linguistic revision, the manuscript was revised by a professional Proof Reading Service (Proofers, GB) before resubmission.

  1. The photograph and scanning electron microscope (SEM) images of the tested materials should be provided to show their morphologies, which might affect the protein adsorption on the materials’ surface.

Answer:           We agree with the reviewer and have made SEM images. A short description of the method, the observations made, and  the knowledge gained from this was integrated into the corresponding parts of the manuscript.

  1. Platelet adhesion is considered as an early event when biomaterials come into contact with blood, and the adhered platelets could induce the following immune responses. I recommend the authors add the platelet adhesion assay on the tested materials.

Answer:          We thank the reviewers for the valuable comment. The final influence of protein adherence and platelet aggregation on the foreign body response has not been clarified. In principle, protein and platelet aggregation assays can of course provide information on this, but this is also not the subject of this study. Rather, the aim here, in this ‘proof of concept’-study is to verify whether a WBSA combined with a proteome profiling array is suitable for the initial characterization of biomaterials. In particular, the simplicity of the implementation is the main focus. If clear differences in the initial foreign body response emerge under these conditions, the concept can be used as a screening tool. The results presented here are encouraging. A corresponding section has been added to the discussion (p.14).

  1. The references related to protein adsorption and platelet-surface interaction on biomaterials should be added and discussed (ACS Appl. Bio Mater. 2020, 3, 5574; Acta Biomaterialia, 2017, 64, 187; Biomaterials, 2010, 31, 1533)

Answer:          We thank the reviewers for the valuable comment. A corresponding section has been added to the discussion (p.12).

  1. To my knowledge, pH 7.4 PBS or HBSS are usually used as buffer solutions in blood biocompatibility evaluation, but in the WBSA experiment the authors used DMEM solutions, a medium for cell culture. Please specify the reason for this.

Answer:           Incubation of the blood with the foreign body was performed over 24h. To maintain cell viability in the blood over this period, DMEM was used instead of a pure buffer solution. This procedure is in accordance with common publications and has already been performed and published by us since the DMEM medium contains also a buffer [1,2][3][4].

  1. De Groote, D.; Zangerle, P.F.; Gevaert, Y.; Fassotte, M.F.; Beguin, Y.; Noizat-Pirenne, F.; Pirenne, J.; Gathy, R.; Lopez, M.; Dehart, I.; et al. Direct stimulation of cytokines (IL-1β, TNF-α, IL-6, IL-2, IFN-γ and GM-CSF) in whole blood. I. Comparison with isolated PBMC stimulation. Cytokine 1992, 4, 239–248.
  2. Russell, M.; Mellkvist-Roos, A.; Mo, J.; Hidi, R. Simple and robust two-step ex vivo whole blood stimulation assay suitable for investigating IL-17 pathway in a clinical laboratory setting. J. Immunol. Methods 2018, 454, 71–75.
  3. Wouters, I.M.; Douwes, J.; Thorne, P.S.; Heederik, D.; Doekes, G. Inter- and intraindividual variation of endotoxin- and β(1 → 3)-glucan-induced cytokine responses in a whole blood assay. Toxicol. Ind. Health 2002, 18, 15–27.
  4. Wutzler, S.; Maier, M.; Lehnert, M.; Henrich, D.; Walcher, F.; Maegele, M.; Laurer, H.; Marzi, I. Suppression and recovery of LPS-stimulated monocyte activity after trauma is correlated with increasing injury severity: A prospective clinical study. J. Trauma - Inj. Infect. Crit. Care 2009, 66, 1273–1280.

  1. Figure 1, please add the description of letter ‘A-J’ and number ‘1-24’ in the figure caption.

Answer:           We agree with the reviewers critism and added a table with cytokines position on array sheet for more clarity.

  1. The information of Figure 4 and Figure 5 are not included in the Results and Discussion, please revise it.

Answer:           We thank the reviewer for the comment. Figure 4 has already been discussed on page 14. Figure 5, however, is an overview of the entire results. They are discussed in detail in the discussion. Nevertheless, we have completely revised Figure 5. We think that now the informative value of the figure has been significantly improved.

Minor

  1. Please keep the range and scale of Y axis consistent in each figure.

Answer:           We revised all figures.

  1. Line 337, revise ‘WBA’ to ‘WBSA’

Answer:           We changed ‘WBA’ to ‘WBSA’

Reviewer 3 Report

Dear Authors, 

I was pleased to review this relevant article.

Summary:

This article introduces novel method how to characterize early immune response to implantable materials. The combination of Whole Blood Stimulation Assay (WBSA) and Protein Profiler Array (PPA) is more complex and therefore more representative compare to simple in vitro culture. On the other hand, less complex, less expensive and mainly less time consuming compare to animal studies. This is a proof of concept study. Investigators showed this concept can be used to characterize the early response and I do hope they will continue to characterize and develop this method further i.e. different time points, different materials.

Minor comments: (please reflect answers to my questions in the text of the article)

1/Why did you choose 24 hours, which time points would you suggest to test further, which longest time-point is based on your knowledge possible?

2/How many specimens per material were tested?

3/How many samples of supernatant were afterward taken for protein analysis?

4/Was it evaluated by one or two investigators? Or the images were clear to evaluate the density?

5/if you use small number of samples – can you clearly show the response of one material is uniform (that all specimens from one material have comparable response) to clarify there are no ,,outliers,,  changing results

6/line 143: This sentence is not clear to me as I hear for first time in the article about group size of 3 and this part,, relatively large comparison groups (n=5),, is not clear to me at all what do you mean by this

Due to the small group size (n=3) and the relatively large comparison groups (n=5), it is also legitimate to consider exploratory statistical significances.

7/line 153: For six proteins, a statistically significant higher release after incubation with PLGA could be determined, in part compared to all other comparison groups (Figure 1). These factors are Gro-1α, IL-5, IL-8, lipocalin, M-CSF and MPO.   – the investigated proteins should be described in method section ( also possible as referring to table)

8/If there are only 3 samples per group I personally would be interested to see results as points on graph, box plot give impression it summarizes tenths of data ( this is connected to my comment 5)

9/There is a missing explanation of abbreviations used in figures (i.e.Fig.2)

General questions:

1/ which host response you consider (based on your knowledge) favorable? it would be for reader clear which proteins he should expect to be expressed in case of favorable reply and which not (which leads to rejection)

2/ Is it for sure, the plasma from different people gives always comparable results, when same material is used? Is there published study which confirms this? (Or could happened if you now take plasma from another 3 people (different from previous) you get different results with same tested materials? Is there some proof of concept how many people you should use to make ,, whole  blood of healthy donors?,,  Do I understand well you made a mixture from those 3 people? Why not one and why not 15 people?

I cite from ref.18: However, even if the deviation of the median within the whole group was very low, it should be noted that in single tests, variations in cytokine release were up to fourfold. After stimulation with 40 pg/mL endotoxin, a confidence interval of 60–140% was calculated for the test repetition. Variance for IL-8 levels was markedly higher than for IL-1β, and a pronounced cytokine release was measurable even without endotoxin stimulation.

Author Response

Reviewer 3

Response to reviewer:

We thank the reviewer for the intensive examination of our manuscript and the constructive criticism. Below we comment on the points of criticism. Since one reviewer requested a linguistic revision, the manuscript was revised by a professional Proof Reading Service (Proofers, GB) before resubmission.

Minor comments

1/Why did you choose 24 hours, which time points would you suggest to test further, which longest time-point is based on your knowledge possible?

Answer:           The aim of the study was to get an impression of the early foreign body reaction. In this phase, mainly the innate immune system is activated by the immune-naive organism (reference). The reaction should thus be basically organism-independent, so that the probability for a material-specific, reproducible result is highest.

From our point of view, therefore, an extension of the observation period does not make sense for the time being. Nevertheless, it would of course be interesting to see whether the material-specific difference detected here increases or decreases over time. This would have to be clarified in a follow-up project.

2/How many specimens per material were tested?

Answer:          Each material was tested once with each blood donation. No duplicate or triple values were determined. However, the spots on the protein profiler array are in duplicate for each parameter.

3/How many samples of supernatant were afterward taken for protein analysis?

Answer:          Only one sample from each supernatant was analyzed. No duplicate or triplicate values were obtained.

4/Was it evaluated by one or two investigators? Or the images were clear to evaluate the density?

Answer:          The chemiluminescence signals were recorded with a fluorescence camera and subsequently analyzed by computer software. The detection of the spots was automated and software-supported for strong to medium spots. Weak spots could not be reliably detected automatically by the software. In this case, the spots were marked manually in the software ImageJ using the polygon tool and the four-eye principle (Two observers). The spot densities were determined for all spots using the corresponding analysis function of the ImageJ software. Thus, intra- and interobserver bias could be avoided.

The following is the corresponding passage from the manuscript:

Membranes were photographed immediately after addition of the chemiluminescence substrate provided with the kit using a Fusion Fx7 gel scanner equipped with a high dynamic range digital camera (Vilber Lourmat, Eberhardzell, Germany). Exposure time was 5 minutes for each membrane. Images were stored as 16 bit non compressed TIFF-files using the software Fusion (Vilber Lourmat).The results were evaluated in a two-step process using the image processing software Image J (http://imagej.nih.gov/ij/). First, strong and medium strong spots were automatically identified and analysed by the software (determination of optical density: size in pixel x mean grey value). The remaining weak spots were finally manually marked and calculated. The manual marking of the spots was checked by a second experimenter.

5/if you use small number of samples – can you clearly show the response of one material is uniform (that all specimens from one material have comparable response) to clarify there are no ,,outliers,,  changing results

Answer:          Thank you for the important note. Only one WBSA per material was performed per donor. In this study, no multiple values per material and donor were measured. Thus, intraindividual variance, as discussed below in the paper, is quite possible. However, interindividual consistency was found for testing one material with three different donors. This also does not suggest a large variance for intraindividual values.

6/line 143: This sentence is not clear to me as I hear for first time in the article about group size of 3 and this part,, relatively large comparison groups (n=5),, is not clear to me at all what do you mean by this

Due to the small group size (n=3) and the relatively large comparison groups (n=5), it is also legitimate to consider exploratory statistical significances.

Answer:          We apologize for this inaccuracy and changed and have added the corresponding paragraph: ‘Statistical evaluation was performed using the Kruskal-Wallis test with Bonferroni-Holm postohoc analysis. P < 0.05 is significant. Due to the small group size (n=3 blood donors) and the relatively large number of comparison groups (n=5 (Control, LPS, PMMA, CS, PLGA), it is also legitimate to consider exploratory statistical significances.’

Furthermore we added a figure for better understanding (Figure 1).

7/line 153: For six proteins, a statistically significant higher release after incubation with PLGA could be determined, in part compared to all other comparison groups (Figure 1). These factors are Gro-1α, IL-5, IL-8, lipocalin, M-CSF and MPO.   – the investigated proteins should be described in method section ( also possible as referring to table)

Answer:          We apologize for this inaccuracy and added the mentioned table.

8/If there are only 3 samples per group I personally would be interested to see results as points on graph, box plot give impression it summarizes tenths of data ( this is connected to my comment 5)

Answer:          We thank the reviewers for this comment. In the boxplots used here with n=3 blood samples/material, all three data points can be clearly identified. Two data points can be read as minimum and maximum, while the third value corresponds to the median (see following figure)

9/There is a missing explanation of abbreviations used in figures (i.e.Fig.2)

 Answer:         An explanation for abbreviations has been added.

General questions:

1/ which host response you consider (based on your knowledge) favorable? it would be for reader clear which proteins he should expect to be expressed in case of favorable reply and which not (which leads to rejection)

Answer:        This is a very valuable indication. The focus of this work is the development of a method for the biomaterials characterization. The present results reveal characteristic protein profiles with material and surface specific differences. Further studies with other biomaterials must now follow to draw concrete conclusions from the results. Too many factors (material, macroscopic and microscopic surface structure, additives, coatings, etc.) can influence the properties. With the current findings, we now want to initiate a broad study to test biomaterials even with only subtle differences, and then investigate the influence of individual factors. Only with such results can the reviewer's question then be answered. Again, it only remains to say at this point that this is a 'proof of concept' study. The intended method should then be able to provide answers to precisely such questions. At the moment it is too early to make such statements.

2/ Is it for sure, the plasma from different people gives always comparable results, when same material is used? Is there published study which confirms this? (Or could happened if you now take plasma from another 3 people (different from previous) you get different results with same tested materials? Is there some proof of concept how many people you should use to make ,, whole  blood of healthy donors?,,  Do I understand well you made a mixture from those 3 people? Why not one and why not 15 people?

I cite from ref.18: However, even if the deviation of the median within the whole group was very low, it should be noted that in single tests, variations in cytokine release were up to fourfold. After stimulation with 40 pg/mL endotoxin, a confidence interval of 60–140% was calculated for the test repetition. Variance for IL-8 levels was markedly higher than for IL-1β, and a pronounced cytokine release was measurable even without endotoxin stimulation.

 Answer:       This is a very important indication that must be addressed in a follow-up study. In this study, no multiple values per material and donor were measured. Thus, intraindividual variance, as discussed below in the paper, is quite possible. However, interindividual consistency was found for testing one material with three different donors. This also does not suggest a large variance for intraindividual values. This was shown by Wouters et al..

The blood samples were not pooled, as this would certainly lead to complications (coagulation, immune reaction, etc.). Increasing the number of donors might well yield more valid results. Here, significant differences are already shown with a donor number of n=3. This is also the aim of an upcoming study.

Round 2

Reviewer 2 Report

The authors have addressed my concerns, and this reviewer would like to recommend its publication.

This manuscript is a resubmission of an earlier submission. The following is a list of the peer review reports and author responses from that submission.